# Validity of the Spanish Version of the Vaccination Attitudes Examination Scale

**DOI:** 10.3390/vaccines9111237

**Published:** 2021-10-24

**Authors:** Borja Paredes, Miguel Ángel Cárdaba, Ubaldo Cuesta, Luz Martinez

**Affiliations:** 1Departamento de Psicología Social y Metodología, Facultad de Psicología, Universidad Autónoma de Madrid, 28049 Madrid, Spain; 2Universidad Villanueva, 28034 Madrid, Spain; mmartincar@villanueva.edu; 3Departamento de Teoría y Análisis de la Comunicación, Facultad de Ciencias de la Información, Universidad Complutense de Madrid, 28040 Madrid, Spain; ucuestac@ucm.es (U.C.); luzmartinez@ucm.es (L.M.)

**Keywords:** vaccinations, scale, individual differences, vaccine hesitancy, validity

## Abstract

Individuals vary in the extent to which they have unfavorable attitudes towards vaccines. The Vaccination Attitudes Examination (VAX) Scale is a recently developed brief 12-item questionnaire created to better understand general vaccination attitudes. The current research aimed at providing a Spanish adaptation of the VAX Scale. After conducting an initial pilot study, Exploratory and Confirmatory Factor Analysis showed that the Spanish version of the scale had good internal consistency and factor structure (Study 1), discriminant validity from other individual differences measures (such as the Beliefs about Medicine Questionnaire and the Medical Mistrust Index) as well as good predictive validity of relevant vaccination-related outcomes (Study 2). In conclusion, in the present research, the Spanish version of the VAX scale proved to have a high internal consistency, showed convergent validity with other conceptually similar constructs, and successfully predicted vaccination intentions and vaccination decisions. Having this scale available in Spanish will allow researchers to analyze vaccination processes and vaccine hesitancy over a great number of people.

## 1. Introduction

Although there is much evidence [1] that vaccines are one of the most effective public health interventions in history, they are still surrounded by criticism [2]. Vaccine rejection has important economic, social and health consequences since wide vaccine acceptance is necessary not only in order to maintain herd immunity and prevent outbreaks of vaccine preventable diseases but also to ensure the adoption of novel vaccines [3]. For example, the long-term success of the public health response to the COVID-19 pandemic will depend on acquired immunity in a sufficient proportion (67% in this case) of the population [4].

Unfortunately, Anti-vaccination attitudes appear to be increasing [5] and might be responsible for the suboptimal uptake of many vaccines [6]. Even in the context of the current COVID-19 pandemic, mistrust in the safety and effectiveness of vaccines globally is widely extended [7]. Despite anti-vaccination movements being mostly based on non-scientific, anecdotal evidence about vaccine side-effects [8,9], their messages have shown to successfully prevent people from getting vaccinated [10,11]. This poses a serious public health issue that experts are focusing on, addressing and solving [12]. Therefore, understanding attitudes underlying vaccine hesitancy or rejection is important not only for predicting behavior but for developing effective interventions as well.

The Vaccination Attitudes Examination (VAX) Scale is a recently developed brief 12-item questionnaire created to better understand general vaccination attitudes [13].

Unlike other scales, the VAX scale does not focus exclusively on specific vaccines or certain demographics groups, thus allowing and facilitating comparisons across different studies.

VAX scale is formed by 4 specific subscales that evaluate: (a) mistrust of vaccine benefit, (b) worries about unforeseen future effects, (c) concerns about commercial profiteering, and (d) preference for natural immunity. Thus, this scale not only provides an efficient method for identifying those with vaccination resistance but also offers an opportunity to better understand the underlying reasons.

Higher total VAX scores suggest stronger antivaccination attitudes. Previous studies have found that the VAX scale is significantly associated with previous vaccination behavior and refusal (e.g., influenza vaccination), as well as intentions to receive future vaccines either for oneself or for one’s children [13]. For example, recent research [14] has shown that the VAX scale was able to predict future intention to receive the COVID-19 vaccine. Additionally, VAX scale scores successfully differentiate parents who vaccinated their children from those who did not. Moreover, VAX scale scores positively correlate with other health relevant measures such as medical mistrust and beliefs about medicines [15].

The aim of this replication study is to determine the validity of the VAX scale in the Spanish language. Although the VAX scale has been translated into other languages [16], a validation of the Spanish version of the VAX scale was still necessary since Spanish is the fourth most widely spoken language in the world with approximately 580 million speakers. On top of that, the Spanish speaking population grew about 30% during the last decade [17]. Regarding vaccine hesitancy in Spanish-speaking regions, Latin America and the Caribbean currently has a lower vaccination rate than the region-wide rates set by the Pan American Health Organization [18], partially due to different forms of vaccine skepticism. Spain has also shown worrying rates of COVID-19 hesitancy [19], even in the absence of a validated measure. Therefore, it is all the more important to have a valid instrument in Spanish that captures general attitudes towards vaccination.

In sum, the present research aims at showing the psychometric and predictive properties of the Spanish version of the VAX scale. Across two studies, participants completed the VAX scale (Study 1) along with other scales aimed at measuring similar constructs and participants’ decisions to vaccinate (Study 2). We predicted that the Spanish VAX scale would replicate the factorial structure of the original scale in English (Study 1) and that it would predict vaccination decisions over and above other competing constructs (Study 2).

## 2. Study 1: Factorial Validity

The main goal of study 1 was to test the factorial validity and the internal consistency of the scale. A pilot study was carried out to run an Exploratory Factor Analysis on the Spanish VAX scale. Then, a study was conducted to test the factorial structure on the new Spanish VAX scale using Confirmatory Factor Analysis. This technique may show the extent to which the different factors loading on to vaccination attitudes in the Spanish VAX scale resemble those of the original scale. This, in turn, would evidence that both measures might be tapping on to the same construct (i.e., factorial validity). Additionally, the internal consistency of the scale was evidenced by testing the Cronbach´s Alpha of each subscale.

### 2.1. Materials and Methods

303 participants (75% female), ranging from 18 to 65 years of age (*M* = 28 years; *SD* = 13) voluntarily completed the Spanish version of the VAX scale. Participants were invited to participate in an individual-differencestudy for undergraduates [Concealed University], their relatives and friends via a snowball technique. After being informed that all collected data would be processed anonymously and confidentially, participants completed the 12-item Vaccination Attitudes Examination scale translated into Spanish, indicating their agreement with each item (1 = Totally Disagree, 7 = Totally Agree). The revised model of the European Federation of Psychologists’ Association (EFPA) for the evaluation of the quality of tests [20] and the established recommendations for successfully adapting measures from one culture to another [21] were followed in order to ensure the accuracy of the final translation.

A pilot study was conducted with the purpose of exploring its factorial structure in the Spanish version. 114 participants completed the Spanish version of the 12-item scale. An Exploratory Factor Analysis was run, using the Pearson correlation matrix, Maximum Likelihood as the estimation method, and Direct Oblimin as the rotation method. Based on the eigenvalues (see Table 1), one factor was explained more than 60 percent of the total variance of the scale. Given this unidimensional result in the Exploratory Factor Analysis, a unidimensional model was also tested in the subsequent Confirmatory Factor Analysis.

To test the internal consistency of the scale, we opted for the most widely used internal consistency index (i.e., Cronbach alpha [22]). Cronbach´s alpha has shown to yield adequate results, so long as a certain number of conditions are met, such as using it on unidimensional scales (or subscales) with few items (for research on the strengths and weaknesses of Cronbach´s alpha, see [23,24,25]).

In order to analyze the factorial structure of the scale, a confirmatory factor analysis was run using AMOS 21.0 [26]. Maximum Likelihood (ML) estimator was used for the item-level CFAs. ML is appropriate for continuous/ordinal data and moderate-to-large sample sizes [27,28]. Two theoretical models were proposed and tested for this scale. Specifically, one model with one latent factor loading onto all items (based on the results of the EFA), and another model with the included four subscales, namely; (1) mistrust of vaccine benefit, (2) worries about unforeseen future effects, (3) concerns about commercial profiteering, and (4) preference for natural immunity, with three items loading onto each factor (for the standardized solutions of both models, see Table 2 and Table 3). This model has been the most used when it comes to explaining the instrument’s latent structure [15,16].

### 2.2. Results

*Internal Consistency.* The twelve-item version test demonstrated good internal consistency in all of its four subscales. (*α* > 0.83). Specifically, mistrust of the vaccine benefit (*α* = 0.90), worries about unforeseen future effects (*α* = 0.74), concerns about commercial profiteering (*α* = 0.86), and preference for natural immunity (*α* = 0.84) showed good internal consistency.

*Factorial Validity.* As Table 4 shows, the four-factor model yields good absolute and relative fit indexes. Specifically, we used the following fit indexes: Chi-square, Normed Fit Index (NFI), Non-Normed Fit Index (NNFI or TLI), Comparative Fit Index (CFI), the Akaike Information Criteria (AIC) and Root Mean Square Error of Approximation (RMSEA). Additionally, an EFA showed that the four factors explained 77.8% of the total variance of the scale.

The lower the values of chi-square, AIC and RMSEA, and the higher the values of NFI, CFI and TLI, the better the model fits the data. Precisely, as a general rule, it can be pointed out that TLI ≥ 0.95, CFI ≥ 0.95 and RMSEA ≤ 0.06 indicate an adequate fit to the data [29]. On the other hand, AIC is a comparative index that penalizes model complexity [30]. Table 2 and Table 3 show the standardized solution for the three models. All weights in the three models were significant (*p* > 0.05), although weights in model 2 are generally higher. Table 5 and Table 6 show areas of low fit in the models. Table 4 shows the fit indices values for both models. These results along with Table 2 and Table 3 suggest that the four-factor model offers an overall robust factorial structure. For this reason, in the present research, we have used the four-factor model.

### 2.3. Discussion

As expected, the Spanish version of the VAX scale yielded very good internal consistency. Cronbach’s alpha values over 0.70 are commonly seen as very good indicators of high internal consistency [31]. Additionally, consistent with Martin & Petrie [13], Confirmatory Factor Analysis showed that a four latent factor model provided the best fit to the data. Secondarily, the evaluation of the scale has also shown that the attitudinal structure found in the Spanish sample is the same as in other populations where the VAX has been used [15,16], supporting its factorial validity (i.e., similar fit between the conceptual and empirical structure in both the original and the Spanish version of the scale) and suggesting its cross-cultural validity (i.e., hinting at the possibility of comparing scores of these two scales across countries and/or cultures). After analyzing the properties of the scale, we moved to examine the extent to which the scale was related to other conceptually related constructs as well as testing the scale´s predictive capability.

## 3. Study 2: Concurrent and Predictive Validity

The main goal of study 2 was to examine the relationship between the Spanish version of the VAX scale and some other constructs that have been recently related to it [32,33]. Following Wood et al., [15] we compared VAX scores with scores in Beliefs about Medicines (BMQ), and Medical Mistrust (MMI). Additionally, this Study was carried out to test the predictive validity of the Spanish VAX scale (compared to the BMQ and MMI) of vaccination intentions as well as vaccination decision-making.

### 3.1. Materials and Methods

283 participants (67.1% female) with ages ranging from 16 to 75 (*M* = 32.63 *SD* = 14.82), were enrolled in the study via an invitation sent to college students [Concealed University], their relatives and friends via a snowball technique. The study was presented as research regarding individual differences, and participants were informed that all data collected for this study would be treated confidentially and anonymously. Once participants were informed that the data would be processed anonymously and confidentially, they completed a questionnaire that included the Spanish version of the VAX scale, the Beliefs about Medicines Questionnaire (BMQ) [32], the Medical Mistrust Index (MMI) [33] as well as a number of dependent measures regarding vaccination intentions and behavior.

### 3.2. Independent Variables

*Vaccination Attitudes Examination (VAX):* Participants completed the 12-item of the VAX scale translated into Spanish (see Appendix A) (α = 0.93). Responses were scored so that higher numbers represented more unfavorable vaccination attitudes.

*Beliefs about Medicines Questionnaire (BMQ)**:* This construct refers to the different ideas people hold about medications, their known and unknown side effects, and potential adherence to prescribed treatments (α = 0.85). Among other reasons, the relationship between the BMQ and the VAX scale was analyzed because both constructs pertain to the beliefs about different forms of medication as a source of impact in one’s behavior. Based on previous evidence [15], it was predicted that there would be a significant relation between these two measures. Responses were scored so that higher numbers represented more negative beliefs about medicines.

*The Medical Mistrust Index (MMI)**:* This 17-item instrument aims at measuring mistrust of health care organizations and examine the relationship between mistrust and health care service underutilization (α = 0.92). We predicted that VAX scores would correlate positively, yet moderately, with MMI because of similar conceptual elements. Responses were scored so that higher numbers represented greater medical mistrust.

*Ancillary Measures:* We included measurements of age, gender, education level (i.e., five options of response; 1 = Primary Education, 2 = Secondary Education, 3 = vocational training, 4 = Undergraduate college degree, 5 = Graduate college degree), and a question asking participants to place themselves in the political spectrum (from 1 = Extreme left-wing, to 7 = Extreme right-wing).

### 3.3. Dependent Variables

*Vaccination Intentions:* Participants were asked to report their intentions towards supporting vaccines on three 7-point (1–7) Likert scales (i.e., “to what extent will you get the flu vaccine in the next season?”, “To what extent will you get the COVID-19 vaccine when it´s available for you”, “To what extent are you willing to sign a manifest defending the widespread use of vaccines?”). These items had high internal consistency (*α* = 0.83), thus were averaged to form an overall vaccination intentions index. Responses were scored so that higher numbers represented greater vaccination intentions.

*COVID-19 Vaccine:* Participants reported their decision to get any of the new COVID-19 vaccines by signing in a box provided by the researchers after the following phrase: “Please sign below if you will get the COVID-19 vaccine when it is made available to you?” (2 = Signed, 1 = Not signed).

### 3.4. Results

A preliminary analysis of the relationships between the continuous variables was conducted using Pearson correlations. As expected, a significant and positive correlation was observed between all variables, *r*(276) > 0.644, *p* < 0.001 (see Table 7).

*Vaccination Intentions:* This dependent variable was submitted to a multiple stepwise regression analysis. VAX, BMQ, MIM, and all the Ancillary Measures were entered as predictors. As expected, the VAX scale was the best predictor of vaccination intentions, *B* = −0.832, *t*(274) = −14.048, *p* < 0.001, followed by MIM, *B* = −0.363, *t*(274) = −2.560, *p* = 0.011, and Age, *B* = −0.363, *t*(274) = −2.560, *p* = 0.011. BMQ and the rest of ancillary measures were excluded from the model, *p >* 0.50. (see Table 8).

*COVID-19 vaccine:* This dependent variable was submitted to a logistic stepwise binary regression analysis. VAX, BMQ, and MIM and all the Ancillary Measures were entered as predictors. As expected, the VAX scale was the best predictor of signing to get the COVID-19 vaccine, *B* = −1.476, *z* = −7.685, *p* < 0.001. Specifically, looking at the odd ratio value [Exp(B) = 0.229], as participants increased 1 point in their Spanish VAX score, their odds of singing to get the COVID-19 vaccine reduced by 77.1%. Political spectrum (i.e., left-right) also remained in the final model as a significant predictor, *B* = −1.476, *z* = −2.294, *p* = 0.021. Specifically, looking at the odd ratio value [Exp(B) = 0.761], as participants moved 1 further into the scale supporting the right-wing (and away from the left-wing), their odds of singing to get the COVID-19 vaccine reduced by 23.9%. All other variables (*B* < 1.3, *p* > 0.250, were excluded from the model. (see Table 9).

### 3.5. Discussion

The present findings replicate previous ones [15], suggesting that the Spanish version of the instrument resembles not only the original structure, but also the relation between the VAX scale and other similar constructs. Additionally, this Study shows that the VAX scale predicts vaccination intentions over and above other similar scales such as the Beliefs about Medicine Questionnaire (BMQ) and the Medical Mistrust Index (MMI). Lastly, and consistent with other findings regarding COVID vaccine hesitancy [34], our data suggests that participants identifying as right-wing were significantly less likely to request and/or accept the COVID-19 vaccine.

## 4. Conclusions

Previous research shows that the VAX scale is a very useful construct for understanding attitudes towards vaccination (not only to identify vaccination resistance and hesitancy but to understand its nature) and predict vaccination behavior [13,14]. All this evidence points to the fact that having a valid and reliable instrument in Spanish that allows to evaluate vaccination attitudes in a simple and effective way would prove very helpful for studying Spanish-speaking populations. Thus, this study assesses the VAX scale in a different and relevant cultural setting through the adaptation of the 12-item VAX scale [13]. furthermore, the analysis showed that the used samples in the VAX scale offered high internal consistency scores. The factorial validity of the scale was tested using Exploratory Factor Analysis and Confirmatory Factor Analysis. The results of our analysis (e.g., Model 2) showed that a four-factor model (relative to a unidimensional model) provides an overall better fit with the data. Similar to the original scale, this indicates that, although the four subscales are correlated, they tap into distinct aspects related to vaccination attitudes. As noted, the use of these subscales might be helpful in increasing our understanding of vaccine hesitancy and its underlying motives. Thus, the Spanish version of the Vax scale makes it possible to discriminate between subjects who have the same score on the general scale but different scores on the subscales. Therefore, this ability to discriminate might be especially helpful for the development of effective strategies in order to improve vaccination attitudes.

Moreover, the results of the second study revealed high correlations between the VAX Scale and other conceptually related constructs (BMQ and MMI), showing good concurrent or convergent validity. Additionally, this study showed that the VAX scale not only was a good predictor of vaccination intention but also was significantly more capable of predicting vaccination decision-making than the other included scales, suggesting discriminant validity. Therefore, these findings suggests that the Spanish version of the VAX scale may be a particularly useful measure to predict future vaccination-related behavior in Spanish-speaking populations.

Researchers interested in applications of this scale in Spanish-speaking populations could explore some socio-demographic moderators. For instance, the literature on vaccine acceptance has shown that, among other factors, female respondents [35], people on the right-wing side of the political spectrum [36], and ethnic minorities [35] tend to show more vaccine hesitancy (for a review on these and other factors on a European population [37]). 

As a possible limitation to the present research, it could be stated that although we used a wide sample across both studies, the greater presence of women and participants under 65 makes it advisable to be cautious when extrapolating the results to the general population. However, the samples used do vary in age, gender, educational background, and political allegiances. This suggests a high external validity within the ranges seen in these variables for the Spanish language. In addition, although the validity of the Spanish version of the Vax scale was stablished across two different samples, our design would not allow for a test-retest analysis (the high variance in attitudes towards vaccinations in Spain at the time of data collection was a clear threat to the potential test-retest reliability results). Therefore, future research would benefit from more comprehensive psychometrics evaluations and temporal consistency scores (i.e., test-retest).

In conclusion, in the present study, the Spanish version of the VAX scale proved to have a high internal consistency, showed convergent validity with other conceptually similar constructs and successfully predicted vaccination intentions. Having this simple, precise, consistent, and valid instrument available in Spanish will allow researchers to analyze vaccination-related phenomena and vaccine hesitancy over a great amount of people as well as a potential starting point for cross-cultural vaccination attitudes studies.

## Figures and Tables

**Table 1 vaccines-09-01237-t001:** Eigenvalues and proportion of explained variance. KMO = 0.922.

Variable	Eigenvalue	Proportion of Variance
1	7.480	0.62330
2	0.968	0.08069
3	0.764	0.06347
4	0.565	0.04712
5	0.444	0.03704
6	0.383	0.02697
7	0.324	0.02521

**Table 2 vaccines-09-01237-t002:** Standardized solution in Model 1.

Item	Model 1
	AV
1	0.78
2	0.75
3	0.58
4	−0.82
5	−0.78
6	−0.82
7	0.39
8	0.68
9	0.58
10	0.62
11	0.75
12	0.76

**Table 3 vaccines-09-01237-t003:** Standardized solutions and correlations between factors in Model 2.

Item	Model 2			
	F1	F2	F3	F4
1	0.91	-	-	-
2	0.78	-	-	-
3	0.92	-	-	-
4	-	0.556	-	-
5	-	0.80	-	-
6	-	0.69	-	-
7	-	-	0.82	-
8	-	-	0.80	-
9	-	-	0.83	-
10	-	-	-	0.68
11	-	-	-	0.87
12	-	-	-	0.84
F1	1	−0.72	−0.83	−0.70
F2		1	0.76	0.71
F3			1	0.84
F4				1

**Table 4 vaccines-09-01237-t004:** Fit Indices from Confirmatory Factor Analysis of translated VAX scale. (Estimation method: Maximum Likelihood).

Model Study 1	χ^2^	*df*	χ^2^*/df*	CFI	NFI	TLI	RMSEA	AIC
1. One factor	397.23	56	7.093	0.85	0.83	0.82	0.14	465.23
2. Four correlated factors	109.78	48	2.287	0.97	0.95	0.96	0.06	193.78
All χ^2^: *p* < 0.01.								

**Table 5 vaccines-09-01237-t005:** Summary Statistics for Standardized Residuals and Largest Modification Indexes for Model 1.

Standardized Residuals	Statistics
Smallest Standardized Residual	−1.568
Median Standardized Residual	0.000
Largest Standardized Residual	1.860
**Error Correlation**	**Modification Index**
E1–E3	111.71
E11–E12	61.08
E4–E6	39.36

**Table 6 vaccines-09-01237-t006:** Summary Statistics for Standardized Residuals and Largest Modification Indexes for Model 2.

Standardized Residuals	Statistics
Smallest Standardized Residual	−1.714
Median Standardized Residual	0.000
Largest Standardized Residual	4.44
**Error Correlation**	**Modification Index**
E4–E6	12.42
E1–E10	10.92
E2–F1	9.89

**Table 7 vaccines-09-01237-t007:** Study 2. Correlations between Vaccination Attitudes Examination, Beliefs about Medicines Questionnaire, The Medical Mistrust Index and Vaccination Intentions. *: *p* < 0.05.

Variables	1	2	3	4	M	SD
1. VAX					3.70	1.52
2. BMQ	0.655 *				2.72	0.75
3. MMI	0.693 *	0.645 *			2.03	0.61
4. Vaccination Intentions	−0.791 *	−0.566 *	−0.608 *		4.45	3.70

**Table 8 vaccines-09-01237-t008:** Study 2. Summary of stepwise linear regression to predict vaccination intentions. *: *p* < 0.05.

	Step 1		Step 2		Step 3	
Measure	β	*t*	β	*t*	β	*t*
Intercept	7.86 *	45.41 *	8.22 *	37.13 *	7.94 *	31.70 *
VAX	−0.92 *	−21.24 *	0.81 *	−13.82 *	−0.83 *	−14.04 *
MMI	-	-	0.36 *	−2.51 *	−0.36 *	−2.56 *
BMQ	-	-	-	-	-	-
Sex	-	-	-	-	-	-
Age	-	-	-	-	0.01 *	2.31 *
Left-Right	-	-	-	-	-	-
Education	-	-	-	-	-	-

**Table 9 vaccines-09-01237-t009:** Study 2. Summary of stepwise logistic binary regression to predict COVID-19 vaccination. *: *p* < 0.05. LCI: Lower Confidence Interval at 95% confidence for *Exp(B)*. HCI: Higher Confidence Interval at 95% confidence for *Exp(B)*.

	**Step 1**			**Step 2**			**Step 3**		
**Measure**	** *Exp(β)* **	** *LCI* **	** *HCI* **	** *Exp(β)* **	** *LCI* **	** *HCI* **	** *Exp(β)* **	** *LCI* **	** *HCI* **
Intercept	1657.8	-	-	1139.9	-	-	1187.9	-	-
VAX	0.228 *	0.146	0.356	0.230 *	0.148	0.357	0.231 *	0.149	0.359
MMI	0.751	0.336	1.680	0.748	0.335	1.672	0.748	0.334	1.671
BMQ	1.213	0.631	2.335	1.213	0.630	2.335	1.203	0.626	2.313
Sex	1.216	0.584	2.533	1.208	0.580	2.513	-	-	-
Age	1.014	0.989	1.040	1.014	0.989	1.040	1.015	0.990	1.041
Left-Right	0.773 *	0.609	0.981	0.774 *	0.610	0.981	0.776 *	0.612	0.982
Education	0.919	0.535	1.588	-	-	-	-	-	-
	**Step 4**			**Step 5**			**Step 6**		
**Measure**	** *Exp(β)* **	** *LCI* **	** *HCI* **	** *Exp(β)* **	** *LCI* **	** *HCI* **	** *Exp(β)* **	** *LCI* **	** *HCI* **
Intercept	1533.7	-	-	1240.1	-	-	1938.7	-	-
VAX	0.240 *	0.158	0.364	0.227 *	0.156	0.330	0.229 *	0.157	0.333
MMI	0.815	0.388	1.713	-	-	-	-	-	-
BMQ	-	-	-	-	-	-	-	-	-
Sex	-	-	-	-	-	-	-	-	-
Age	1.015	0.990	1.040	1.015	0.990	1.040	-	-	-
Left-Right	0.766 *	0.607	0.967	0.766 *	0.607	0.965	0.761 *	0.603	0.960
Education	-	-	-	-	-	-	-	-	-

## Data Availability

The data presented in this study are openly available in The Open Science Framework (OSF) at doi: 10.17605/OSF.IO/9PUJK.

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
