# Peer review of "Validity of the Spanish Version of the Vaccination Attitudes Examination Scale"

_vaccines, 2021, doi:10.3390/vaccines9111237_

Round 1
Reviewer 1 Report
The authors used cross-sectional studies to address the validity and reliability for Spanish version of the Vaccination attitudes examination scale. Because vaccine hesitancy became one of the biggest public health problems, it would be meaningful to identify the validity and reliability of the scales. However, this article has not been fully answered some of questions due to the insufficient description and inadequate statistical analysis.
First, authors claimed that they confirmed the validity of scale, but they only examined the correlation of each subscale and the predictive values of the subscales in this study. In order to examine the validity, it is necessary to confirm three validities (i.e., conceptual validity, internal validity and external validity), but they have not done so. Similarly, for reliability, they have NOT confirmed it using repeated measurements (i.e., internal validity).
Second, in this paper, the description that should be written in the method section and the description that should be written in the result section are mixed. For example, the method section of Study 1 does NOT describe what kind of statistical method is used to conform the reliability, and the results of statistical analysis suddenly appear in the result section. The description should be organized as it makes it difficult for readers to understand.
Finally, they did NOT use adequate statistical technique in their analysis. For example, they used logistic binary regression in table 9, but they did NOT show the odds ratio. Moreover, as these results may be affected by potential confounding factors such as age and gender, they should include the potential confounding factors in their models.
Author Response
Response to Reviewer 1 Comments
Point 1: The authors used cross-sectional studies to address the validity and reliability for Spanish version of the Vaccination attitudes examination scale. Because vaccine hesitancy became one of the biggest public health problems, it would be meaningful to identify the validity and reliability of the scales. However, this article has not been fully answered some of questions due to the insufficient description and inadequate statistical analysis. First, authors claimed that they confirmed the validity of scale, but they only examined the correlation of each subscale and the predictive values of the subscales in this study. In order to examine the validity, it is necessary to confirm three validities (i.e., conceptual validity, internal validity and external validity), but they have not done so. Similarly, for reliability, they have NOT confirmed it using repeated measurements (i.e., internal validity).
Response 1: We appreciate this point, as it might have been the case that we unpacked certain aspects of validity a bit too lightly in the original manuscript. In the revised manuscript, we have added further explanations on the different aspects of validity and reliability supported by our data. In addition to this, below you will find our rationale for the general absence of the aspects of validity mentioned by reviewer 1 (i.e., conceptual, internal, and external validity):
Content (or conceptual) validity refers to the “adequacy with which a measure assesses the domain of interest”. This is a key feature of any validated measure, hence the adequacy of content is vital if the items are to measure what they are presumed to measure (DeVellis, 2012). The reason why this form of validity is not directly covered in this manuscript is because these aspects of content validity and adequacy were thoroughly covered by the original creators of the scale (Martin & Petrie, 2017). To the extent that factorial validity, predictive validity, and concurrent validity are guaranteed in the translated scale (i.e., we provide supporting data in the manuscript), content validity is relied upon the original scale. Nevertheless, we have decided to include the following clarification in the discussion of the first study.
“As expected, the Spanish version of the VAX scale yielded very good internal consistency. Cronbach’s alpha values over .70 are commonly seen as very good indicators of high reliability (Cortina, 1993). Additionally, consistent with Martin & Petrie [13], Confirmatory Factor Analysis showed that a four latent factor model provided the best fit to the data. Secondarily, the evaluation of the scale has also shown that the attitudinal structure found in the Spanish sample is the same as in other populations where the VAX has been used [15,16], supporting its factorial validity (i.e., similar fit between the conceptual and empirical structure in both the original and the Spanish version of the scale) and suggesting its cross-cultural validity (i.e., hinting at the possibility of comparing scores of these two scales across countries and/or cultures). After analyzing the properties of the scale, we moved to examine the extent to which the scale was related to other conceptually related constructs.”
Internal validity refers to the degree to which a design controls extraneous variables and, thus, permits causal inferences to be made regarding the association between the independent and dependent variable (Flanelly et al., 2018; Campbell & Stanley, 1966). Given that most scale validations take place in correlational designs (i.e., high degree of extraneous variables and low degree of control), internal validity is often overlooked in this context. However, statistical control over certain potentially confounding variables (i.e., age, gender, socio-economic status, political affiliation, etc.) can and should be provided. That is why we have incorporated these variables into the models in Study 2 to test the extent to which they can predict vaccination-related behavior over and above the VAX scale. Lastly, we have added a more detailed explanation of the internal consistency analysis.
External validity refers to the generalizability of the effects found in a given research to other constructs, contexts and/or populations (Campbell., 1957). Although we can only speculate about the actual generalizability of the current set of results, concurrent and discriminant validity are often cited as evidence for external validity (Robinson et al., 2016). In that regard, our manuscript offers samples that vary in age, gender, educational background, and political allegiances. We understand that this point may have not been apparent enough in the original manuscript and we have corrected it in the revised version.
Reviewer 1 also mentioned the lack of a temporal consistency measure (i.e., test-retest).
We agree with reviewer 1 in this being a limitation of the manuscript, therefore we mentioned it in the original manuscript. We have decided to unpack our rationale for this absence more explicitly in the revised manuscript. Specifically, the shift in attitudes towards vaccinations in Spain at the time of data collection was a clear threat to the potential test-retest reliability results.
Point 2: Second, in this paper, the description that should be written in the method section and the description that should be written in the result section are mixed. For example, the method section of Study 1 does NOT describe what kind of statistical method is used to conform the reliability, and the results of statistical analysis suddenly appear in the result section. The description should be organized as it makes it difficult for readers to understand.
Response 2: We have moved those paragraphs to the correct section in the revised manuscript.
Point 3: Finally, they did NOT use adequate statistical technique in their analysis. For example, they used logistic binary regression in table 9, but they did NOT show the odds ratio. Moreover, as these results may be affected by potential confounding factors such as age and gender, they should include the potential confounding factors in their models.
Response 3: We have unpacked the internal consistency analysis for our first study, included the proportion of explained variance for each factor in Study 2, incorporated the χ2/df values for the confirmatory factor analysis, and we have provided the odd ratios for the logistic binary regression in Study 2. We understand that all these inclusions have modified the tables significantly. Therefore, we are happy to remove, add and/or trim whatever information you and the editors suggest.
References
Campbell, D. T. (1957). Factors relevant to the validity of experiments in social settings. Psychological Bulletin, 54(4), 297–312. doi:10.1037/h0040950
Campbell, D. T., & Stanley, J. C. (1966). Experimental and quasi-experimental designs for research. Chicago, IL: Rand McNally & Company.
DeVellis RF. Scale Development: Theory and Application. Los Angeles, CA: Sage Publications (2012).
Flannelly, K. J., Flannelly, L. T., & Jankowski, K. R. (2018). Threats to the internal validity of experimental and quasi-experimental research in healthcare. Journal of health care chaplaincy, 24(3), 107-130.
Martin, L. R., & Petrie, K. J. (2017). Understanding the dimensions of anti-vaccination attitudes: The vaccination attitudes examination (VAX) scale. Annals of Behavioral Medicine, 51(5), 652-660.
Paredes, B., Stavraki, M., Díaz, D., Gandarillas, B., & Briñol, P. (2015). Validity and reliability of the Spanish version of the Revised Self-Monitoring Scale. The Spanish journal of psychology, 18.
Robinson, S., Kissane, D. W., Brooker, J., Hempton, C., Michael, N., Fischer, J., ... & Burney, S. (2016). Refinement and revalidation of the demoralization scale: The DS‐II—external validity. Cancer, 122(14), 2260-2267.

Reviewer 2 Report
This paper is well structured. However, I would like the results of the construct reliability, as well as the convergent validity (average variance extracted) for each of the four factors of the scale. Both construct reliability and convergent validity are extremely important when we want to publish the adaptation of a scale. As the authors performed a confirmatory factor analysis, it will be very easy for them to add this data. In the table with the adjustment indices from the confirmatory factor analysis, it would be easier to read and interpret the χ2/df value instead of placing the values separately, as the reader will have to perform this calculation to check which is the most adequate value between the two models.
Author Response
Point 1: This paper is well structured. However, I would like the results of the construct reliability, as well as the convergent validity (average variance extracted) for each of the four factors of the scale. Both construct reliability and convergent validity are extremely important when we want to publish the adaptation of a scale. As the authors performed a confirmatory factor analysis, it will be very easy for them to add this data. In the table with the adjustment indices from the confirmatory factor analysis, it would be easier to read and interpret the χ2/df value instead of placing the values separately, as the reader will have to perform this calculation to check which is the most adequate value between the two models.
Response 1: We have unpacked the internal consistency analysis for our first study, included the proportion of explained variance for each factor in Study 2, incorporated the χ2/df values for the confirmatory factor analysis, and we have provided the odd ratios for the logistic binary regression in Study 2. We understand that all these inclusions have modified the tables significantly. Therefore, we are happy to remove, add and/or trim whatever information you and the editors suggest.

Round 2
Reviewer 1 Report
The authors revised the manuscript. However, this article has not been fully answered some of questions due to the insufficient description.
First, as I mentioned in the first review, this article does NOT examine the reliability (i.e. by test and re-test). I strongly recommend to delete the word "reliability" in title, abstract (L15), introduction (L62, L83), materials and method (L112), results (L133), and conclusions (L323)
Second, I could NOT understand why authors need to add values of β(EXP) in table 8. I recommend to delete the values of β(EXP) in table 8, if authors did NOT include the β(EXP) in their models.
Finally, authors showed the values of β, S.E., wald, in table 9, but odds ratio and 95% confidence interval would be more suitable to understand for readers. I recommend to show the values of odds ratio and 95% confidence interval instead of those of β, S.E., wald, in table 9.
Author Response
Point 1: First, as I mentioned in the first review, this article does NOT examine the reliability (i.e. by test and re-test). I strongly recommend to delete the word "reliability" in title, abstract (L15), introduction (L62, L83), materials and method (L112), results (L133), and conclusions (L323)
Response 1: Given the lack of any test re-test data on the scale, we have followed the suggestion of reviewer 1 of deleting any reference to the word “reliability” throughout the manuscript (i.e., in title, abstract, keywords, introduction, materials and methods, results, and conclusions). We also acknowledge this lack of a temporal consistency measure as a limitation on this revised version of the manuscript. In some instances, we have substituted the word “reliability” for the “internal consistency”. In other instances, we have removed the mention of reliability altogether.
Point 2: Second, I could NOT understand why authors need to add values of β(EXP) in table 8. I recommend to delete the values of β(EXP) in table 8, if authors did NOT include the β(EXP) in their models.
Authors showed the values of β, S.E., wald, in table 9, but odds ratio and 95% confidence interval would be more suitable to understand for readers. I recommend to show the values of odds ratio and 95% confidence interval instead of those of β, S.E., wald, in table 9.
Response 2: Reviewer 1 also suggested that certain statistics be removed from some tables [e.g., Exp(β) from Table 8 and β, S.E., wald, from table 9) and others be added (odds ratio and 95% confidence interval). We appreciate this suggestion and we have decided to modify the tables accordingly.

Round 3
Reviewer 1 Report
Authors revised their manuscript and it is considered to have reached the quality of publication.